# The Gender Perspective of Professional Competencies in Industrial Engineering Studies

**Maria Inmaculada Tazo** [1]**, Ana Boyano** [2]**, Unai Fernandez-Gámiz** [3],*  and
**Amaia Calleja-Ochoa** [2],*

[1]   Department of Thermal Engines and Machines, Faculty of Engineering of Vitoria-Gasteiz, University of the Basque Country UPV/EHU, 01006 Vitoria-Gasteiz, Spain; mariainmaculada.tazo@ehu.eus

[2]   Department of Mechanical Engineering, Faculty of Engineering of Vitoria-Gasteiz, University of the Basque Country UPV/EHU, 01006 Vitoria-Gasteiz, Spain; ana.boyano@ehu.eus

[3]   Nuclear Engineering and Fluid Mechanics Department, Faculty of Engineering of Vitoria-Gasteiz, University of the Basque Country UPV/EHU, 01006 Vitoria-Gasteiz, Spain

*   Correspondence: unai.fernandez@ehu.eus (U.F.-G.); amaia.calleja@ehu.eus (A.C.-O.)

**Abstract:** Sexism and gender relations in higher education require special attention and are a topic of great interest in regulations related to education. The low participation percentage of women in Science, Technology, Engineering, and Mathematics (STEM) studies has been identified as one of the main problems that must be resolved in order to close the gender gap that exists in the technology sector. The purpose of this study was to investigate the influence of professional competences on the selection of university studies according to the absence or presence of masculinization factors in those studies. Mechanical engineering competences, both generic and transversal, and competence acquisition methods, are classified into 'care' (feminine) or 'provisions' (masculine) concepts. After the competence analysis, it can be concluded that explicit engineering curricula are focused on "provisions", which translates into a cultural perception of industrial engineering as a male profession. After a professional competence analysis in engineering studies at The University of the Basque Country (UPV/EHU), our study identified a relationship between the masculinization factors included in professional competences and the selection of university studies. This paper presents working actions towards the incorporation of a gender perspective into the degree in mechanical engineering at the UPV/EHU.

**Keywords:** gender; university; professional competences; engineering

## 1. Introduction

"Gender perspectives" can be defined as the study of male and female cultural and social constructions, which identify feminine and the masculine characteristics. This socio–cultural analysis is focused on exploring the power inequalities between men and women [1].

In recent years, the study of the impact of sexism and gender relations on university education has been a topic of great interest all around the world. Many studies have shown that women remain greatly underrepresented in top positions of many institutions, even at university. As a sample, some recent studies are commented below. In 2016, one study showed that in the Italian academia, especially in top positions, women are underrepresented; it has been found that when a small number of positions is available, females have a significant lower probability of promotion, especially for full professor promotion [2]. According to the U.S Department of education, even if between 2006 and 2017 women presence in doctoral degrees overpassed 50%, fewer women are found in the highest academic rank positions [3]. In 2019, the analysis of the gender pay gap among UK academic economists concluded

that job rank is found to be a determinant factor, in other words, women are paid less because they cannot reach the highest positions [4]. The same conclusion has been reached after analyzing the University of Valencia in Spain in 2015 [5].

In fact, one of the foundations of the European Higher Education Area (EHEA) is to develop Europe; since one of the 'Millennium Development Goals' is "Achieving gender equality and empowering all women and girls", it is considered necessary to promote studies and research introducing a gender perspective [6].

From a legislative point of view, the Spanish Organic Law 6/2001 (21/12/2001) of Universities has been modified by the Organic Law 4/2007 (12/04/2007). The preface to this law states that "the University's role as an essential transmitter of values . . . is to achieve a tolerant and egalitarian society, in which the fundamental rights, freedoms, and equality between men and women are respected" (Preamble paragraph 12) [7]. Moreover, law 14/2011 (01/06/2011) on Science, Technology, and Innovation considers "the incorporation of the gender approach with a transversal character" important for the development of science (Preamble) [8].

The low participation of women in Science, Technology, Engineering (all engineering degrees), and Mathematics (STEM) studies is identified as one of the main problems that must be resolved in order to close the gender gap that exists in the technology sector [9,10]. Although women have made significant progress regarding their presence in technical studies, they are still under-represented in these fields. Some research has concluded that women in engineering express less confidence in their STEM skills than men do [11–13]).

When the latest data and numbers on the Spanish University System (published by the Ministry of Education, Culture, and Sports) corresponding to the 2015–2016 academic year [14] are analyzed, 54.1% of total university students are shown to be women. Women are the majority in all fields, with the exception of technical degrees. In Engineering and Architecture studies, in the last 10 years, women's presence has suffered a reduction of 2.7 points, from 27.7% in the course 2007/2008 to 25.0% in the academic course 2017/2018 [14]. This reduction in women's presence in this last 10 years has suffered little change and has maintained stable. The highest percentage of women, 69.4%, is in the Health Sciences, and the lowest, 25.8%, is found in the fields of engineering and architecture. These figures have remained stable over the past 10 years. According to the She Figures report (2018) [15], the share of women among academic staff in the EU rapidly declines as they advance to higher positions in research organizations. In 2016, women made up 46% of Grade C staff, defined as the first grade, 40% of grade B staff, and 24% of grade A staff.

On the other hand, according to some recent research [16,17], young people believe that there is now equality between men and women. This phenomenon is called the 'equality mirage', which explains how equality is perceived by society, albeit differently from the real situation. Therefore, it is essential to facilitate interest in gender topics in the university environment. Various studies on the subject note the absence of females in engineering and technology and the fact that social gender stereotypes are evident in engineering, which is considered as a male profession [18–21]. Indeed, it is commonplace to consider engineering as something masculine. In a study conducted in the University of Gävle, Gävle, Sweden [22], in an attempt to analyze the influence of gender in Sustainability Learning, the authors observe that, traditionally, engineering and nursing have been dominated by one gender (the former by males and the latter by females). Indeed, in 2013, across all Swedish universities, women were awarded 28 per cent of the degrees in Engineering and 86 per cent of the degrees of the nursing degrees.

Moreover, gender discrimination persists in the pursuit of academic careers. Recent studies confirm the low representation of women in academic careers at the university level, even though there is parity of postgraduation degrees between women and men [23]. The discrimination suffered by women in this field is not explainable by poorer scientific productivity in the established promotional system [24]. At the university, the upper levels of the hierarchy are mostly male, and horizontal segregation is present. Women remain concentrated in the fields of teaching and health sciences. This vertical segregation correlates with the different levels of responsibility in the hierarchy, both in academia and in the governing bodies that occupy them [25]. The results obtained from a survey

conducted in 2013, at the 24 Russell Group universities in the UK [4], show that women tend to progress at a slower rate than men. The authors called this "the gender effect", wherein a man is more likely to have a higher academic rank due to being a man.

Indeed, according to the National Statistics Institute [26], during 2010–2011 at Spanish public universities, among 10,321 Professors, only 1865 were women, with half of the presented theses read by women in that course. Moreover, discrimination due to sex at university also occurs in subtle ways, such as language or interpersonal relationships [27].

General and worldwide university organizational culture contributes to the transmission of guidelines on gender relations by means of the presence or absence of a gender perspective in a school's curricular content, teaching materials, or research [28]. It is accepted that there are two types of curricula [29]: an explicit one and a hidden one. The former pertains to study plans, learning methodologies, outcomes, competences, evaluation systems, etc. [30]. The latter reflects sets of thoughts, assessments, and beliefs that inform the relationships and practices between people [31,32].

The analysis of a gender perspective for professional competencies at university usually relates to the explicit curricula. A hidden curriculum study will require student and teacher interviews to determine their thoughts on the relevant professional competences. Therefore, we carry out an androcentric value analysis in our objectives, as well as in the competences and methodologies. Moreover, the use of a generic "masculine" identity, both in the formal discourse and in the discourse of the teaching staff, is required in order to achieve an integral formation that includes competences such as collaborative work, critical thinking (feminine), etc. The study and analysis of gender in the higher education system and, in particular, in largely masculinized areas, is necessary if we are to eliminate any form of discrimination against women, as included in the European Pact for Gender Equality [33]. For this, the contribution of the university in these areas is essential to help people contribute to this transformation and become aware of the situation and the consequences.

New learning methodologies are based on the description and implementation of activities that are applied to the resolution of problems related to the required professional profiles. In the development of the professional activity of engineering, there is contact between people, either with the final recipients (e.g., via house heating, machine protection, vehicle retention systems, etc.) or because people intervene in the development of an activity (decision-making, responsibilities, contacts, etc.). People–people relations include gender relations, domination/subordination, and conditioning by socio–cultural values. In the same way that the transversal competences of decision-making are developed, gender equality and social responsibility competences should also be studied at university [34].

Frequently, gender studies in the education sector have focused on gender distribution data [35,36], study selection factors [37,38], interviews with students about their feelings [39,40], working in mixed groups, etc. The professional competence analysis of university degrees [41–43] is also an important factor that determines study selection. In this case, the motivation comes from the fact that despite the efforts made to increase the number of enrolled women, it is observed that the number of enrolled female students is decreasing. Therefore, it was decided to analyze the culture of mechanical engineering, since it is one of the degrees with the lowest presence of women and it is the degree in which the authors teach, starting with the analysis of the competencies that graduates must acquire. However, since engineering degrees are defined through professional competences, the present work presents a professional competence analysis in engineering studies at The University of the Basque Country (UPV/EHU) in order to determine the influence of professional competences on the selection of university studies according to the absence or presence of masculinization factors in a chosen field of study. As it is not possible to modify university degrees' professional competences, several actions are proposed for the inclusion of a gender perspective in the degree in mechanical engineering using active methodologies for the development of 'care' skills. Initially, competences (both generic and transversal) and competence acquisition methods are classified into 'care' (feminine) or 'provision' (masculine). The concepts of 'care' and 'provision' were originally developed to analyze, from a feminist point of view, domestic production and the sexual division of labor, which constitute very important aspects in the field of

gender studies [44,45]. A study performed within the functions of vice-rectors from forty-eight public Spanish universities present gendered divisions of labor among feminized (attention to students and administrative/social/cultural functions), masculinized (strategic research/innovation/technologies), and neutral functions [46]. These concepts are analogous to the concepts of technical engineering skills and professional skills reported by engineers involved with engineering services [47] for engineering community-based challenges. These concepts will be taken as the basis for analyzing and classifying the transversal competences and the teaching–learning methodology.

### 1.1. Background

#### 1.1.1. Professional Competences and Gender

"Professional profiles", can be understood as a "set of knowledge, techniques, skills, and attitudes that are demanded for a certain job"—that is, the characteristics that a job candidate must meet. Professional profiles integrate "professional competences" that include the sets of knowledge, skills, and attitudes linked to particular profession roles, functions, and activities [48]. Professional competences can be divided into three different levels depending on the competence type:

- *Basic competences*: Derived from schooling (reading, writing, comprehension, communication, etc.) from application to performance (planning, reasoning, problem solving, organization, etc.)
- *Generic competences*: Derived from work activities present in several domains (transferable).
- *Technical competences*: Derived from specific occupational actions.

There is also a relationship between competences and employment factors. Generic competences include factors such as age, sex, race, belonging to an ethnic minority, health status, socioeconomic situation, family configuration, migrant status, residence, and transport community support networks. Job performance competences are related to basic skills, centrality, employment, motivation, job search, relational skills, personal assessment, conciliation with workloads, and support networks. professional academic competences include regulated training, complementary training, licenses and certificates, professional experience, generic and technical skills, time availability, displacement availability, availability of contracting modality, and updates in the market.

In this sense, socio–labor gender stereotypes and the explicit or implicit rules according to these stereotypes give rise to biased professional roles and influence the professional competences demanded by women and men. Figure 1 [42] shows examples of socio–labor gender stereotypes and professional roles.

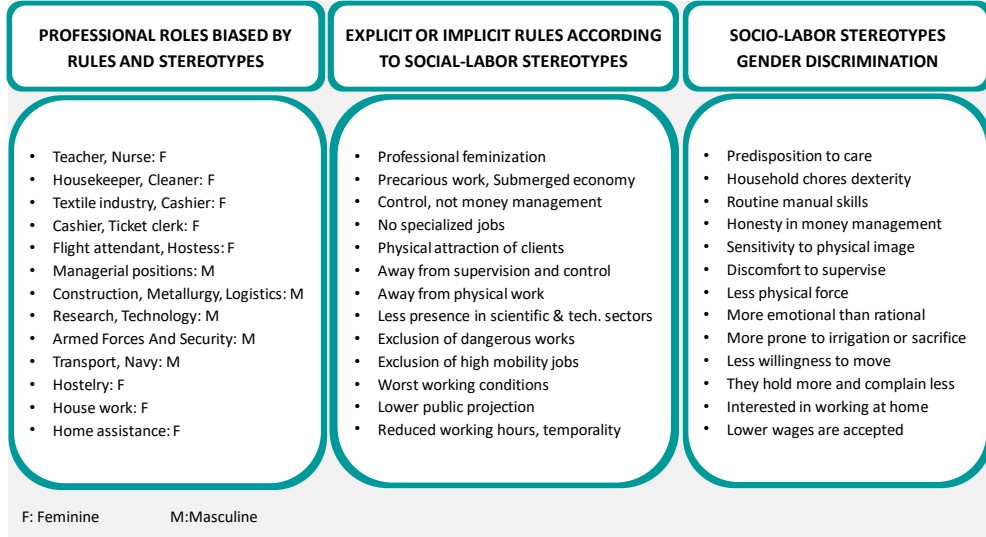

**Figure 1.** Socio-labor stereotypes and professional roles.

### 1.1.2. Professional Competences in University System

The analysis of gender performance and the introduction of a gender perspective in university teaching entail reflecting on all the elements, subjects, and aspects involved in this process in order to identify possible gender biases and eliminate them [49].

There is an influence of gender on university system competences. The MECD (Ministry of Education, Culture and Sport) [50] defines competences as "A combination of knowledge, skills (intellectual, manual, social, etc.), attitudes and values that will enable a graduate to deal with guarantees with the resolution of problems or intervention in a subject in an academic, professional or social context". Gender concepts and values are outlined when referring to cultural/social values such as abilities, attitudes, and psychological resources. These values (dependent on social culture, which also depends on the history, financial profile, and area in which the university degree was developed) are different in each country and will pertain to feminine/masculine elements, commonly prioritizing the values associated with the male gender over those associated with the female gender. In addition, gender is also present in teaching methods and learning activities due to people–people relationships.

Moreover, the term 'competences' in the European university system is linked to the process of the transparent harmonization of degrees and directly links university education with the professional world and with the free movement thereof. In the United States, there has also been a transformation in engineering education towards a competency-based pedagogy [51]. There are at least three levels of narrow and relevant factors: the formulation of degrees, the use of methods, activities, and learning resources, and evaluations. The OCDE (organization for economic cooperation and development) project section, 'Competences definition and selection' [52], notes that competence entails more than knowledge and skills. It includes the ability to face complex demands by putting into action, in specific situations, psychological resources, skills, and attitudes [53].

University degrees are defined by professional generic competences (transversal competences) and by professional specific competences. On one hand, generic professional competences, as seen in Figure 2, can be divided according to the following classification:

- *Instrumental competences*: Those with an instrumental function. They can be simultaneously cognitive, methodological, technological, and linguistic.
- *Interpersonal competences*: Those related to social interaction and communication.
- *Systemic competences*: Those related to comprehension, sensibility, and knowledge. They represent how individual parts can be related and grouped.

Commonly, instrumental competences are considered masculine competences similar to masculine stereotypes and instrumentalism. Interpersonal competences are considered similar to feminine stereotypes, due to their focus on communication and social abilities. Systemic competences are neutral competences because they are related to knowledge necessity (masculine) and sensibility (feminine).

On the other hand, professional specific competences are divided into three different groups: academic competences related to knowledge (know); disciplinary or practical knowledge sets required for each professional sector (do); finally, professional competences, including the communication and inquiry abilities applied to the exercise of a specific profession (know-how) [53].

For professional competences and formation activities, 'care' (feminine) and 'provision' (masculine) concepts can be defined (Figure 3), depending on the presence or absence of the relationship between people and things.

Taking into account this previous classification, professional competences can be described as being related to 'care' and 'provision' concepts (Table 1).

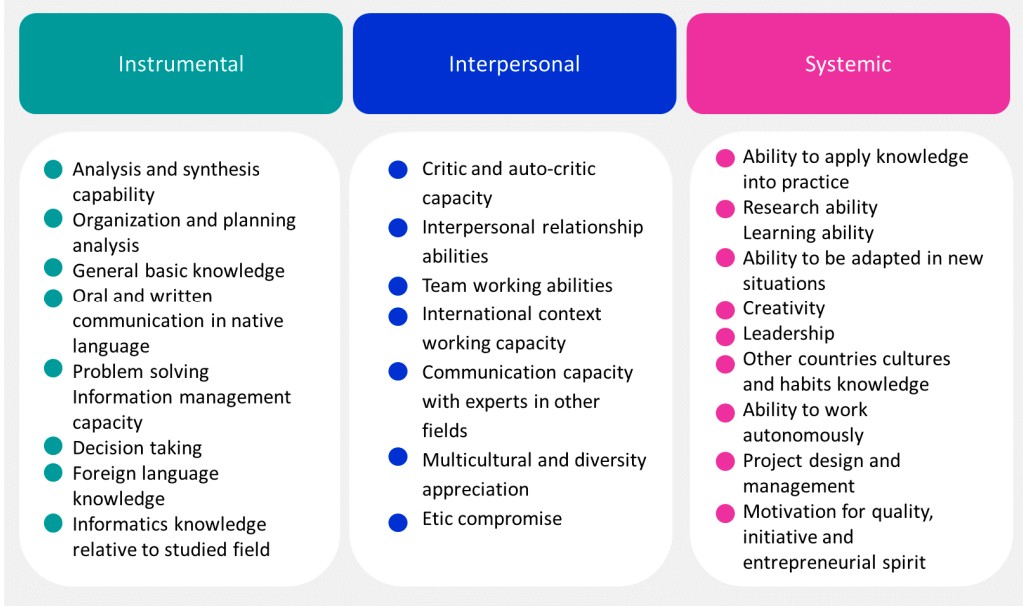

**Figure 2.** Various professional competences classified into instrumental, interpersonal, and systemic competences.

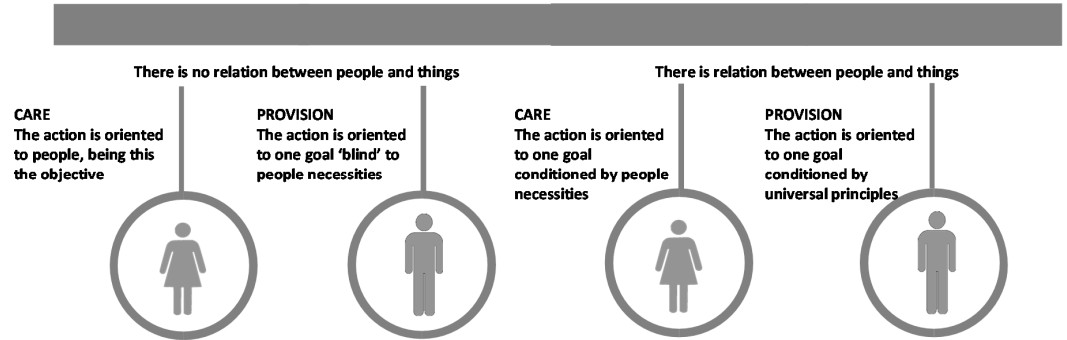

**Figure 3.** Professional competences and formation activity classification according to the type of relationship.

**Table 1.** Professional competence classification.

| | Type | Description | Care/Provision |
|---|---|---|---|
| **Transversal** | Interpersonal | Teamwork and relationship related abilities | Care |
| | Systemic Opening Personal | Systemic competences refer to personal characteristics related to individual behaviour inside a complete system. In 'opening' cases, the receiver position is similar to a passive attitude | Care |
| | Systemic Impact Personal | Personal attitude related to system incidents | Provision |
| **Specific** | Instrumental Cognitive | Related to comprehension, analysis, synthesis, reasoning, and idea-organising activities | Provision (activities related to resources or environmental control) |
| | Instrumental Technical-Scientific | Concrete scientific or technic knowledge development ability | Provision |
| | Professional interest | Vocational aspects such as motivation or interest in certain degrees and knowledge | Provision |
| **Generic** | Instrumental Linguistic | Oral or written communication ability or language knowledge | Care (activities considering attention to other people) |
| | Instrumental Practical | Linked to organizing, planning, and management abilities | Provision |

(Competence Type)

### 1.1.3. Professional Competences in Work Environment

The term 'employability' refers to the ability of a person to be employed in a position offered by the labor market. Some people have greater employability than others; this factor depends on the individual as well as the relevant companies and market trends [53].

Although employability is one of the objectives to be achieved with new university degrees, a gender wage gap still exists, and there are several factors that need to be taken into account. There is also a gap in employability that affects not only the knowledge or qualifications that a person may have but also their skills and networks of contacts. To achieve professional aspirations, and thus an acceptable degree of employability, personal variables need to be considered, such as knowledge, skills, attitudes, experience, personal brand, and values, as well as external variables, such as the socio-economic situation's influence on the gender system.

Professional competences are defined and mainly linked to professional formation in its three levels: the regulated professional training, the occupational professional training (labor reinsertion) and continuous formation. Table 2 shows professional competence classification.

**Table 2.** Hay McBer competence summary [54].

| Conglomerate | Competencies |
|---|---|
| I. Achievement and Action | • Achievement Orientation<br>• Order concern, Quality, and precision<br>• Initiative<br>• Information Search |
| II. Support and Human Service | • Interpersonal understanding<br>• Customer Service Orientation |
| III. Impact and Influence | • Impact and Influence<br>• Organizational Awareness<br>• Establishment of Relations |
| IV. Management | • Development of Others<br>• Assertiveness and Use of Positional Power<br>• Teamwork and Cooperation<br>• Team Leadership |
| V. Cognitive | • Analytical thinking<br>• Conceptual Thinking<br>• Expertise |
| VI. Personal Effectiveness | • Self-control<br>• Self-confidence<br>• Flexibility<br>• Organizational Commitment |

### 1.1.4. Employability and Gender

Graduate employability is one of the most fundamental factors for university degree selection, and it is also essential for the accreditation agencies of teaching and university management. On the

other hand, in the labor insertion report, in addition to employment rates, graduates' satisfaction with their degree and the level of their acquired competences are considered to be transversal and fundamental for the labor world and are also analyzed. In this sense, gender influences professional competences and, as a consequence, employability.

It can be concluded that employability is not a monolithic construct but is instead conditioned by personal factors (gender, ethnic origin, or age) and socio-economic factors at specific historical moments, causing occupational imbalances between women and men. Therefore, employability is a quality or competence of the individual, and, therefore, the responsibility to enter or stay in the labor force falls on the subject. Results reveal that female occupational representation is still characterized by structural factors. The possibility of a female person's insertion into the labor force does not exclusively depend on taking advantage of the possibilities offered by educational and productive agents; other factors are also influential. Therefore, dimensions are defined to stablish gender dimension in the framework of employability. In this sense, two dimensions are defined, interrelated, to position ourselves in the concept of employability: Dimension I. Gender factor and other structuring factors of personal identity. Dimension II. Deficiencies of the socioeconomic system. Based on these dimensions, we interpret that employability is not a monolithic construct but varies according to the factors [55].

Dimension I.

1.  Single-parent mothers and those with new families. In the labor context, these women have limitations in accessing jobs that require high qualifications and deep dedication.
2.  The role of care attributed to women. The sexual division of labor and the ideology that sustains it gives prestige to the role of women in reproduction and care.
3.  Double working day. The domestic tasks and types of care that women develop determine that their jobs outside the home are generally considered salaries to support the domestic economy.
4.  Horizontal labor segregation. The entails an undervaluation of professions and occupational sectors historically represented by females.
5.  Vertical labor segregation. Women gradually disappear in the occupational hierarchy the closer they approach management positions.
6.  Public policies articulated against gender equality. Current public policies provide economic incentives for traditional families to be maintained.
7.  Social class and ethnic origin.

Dimension II. The collateral damage of the crisis of women is manifested in the reinforcement of stereotypes. Commonly, such situations of economic deterioration multiply social discourses that carry a burden of social discrimination, especially gender.

Given both the structural and personal dimensions involved in employability, the key competences involved in employability will have to be considered when analyzing their influence on women's employability, primarily to adapt to the social and labor context and to identify employment-demanding occupations.

## 2. Method

We collected data on the percentages of women and men enrolled in the Spanish university system and their distribution, taking into account the relevant area of knowledge. We also analyzed female enrolment depending on the knowledge area. Afterwards, professional competences in engineering degrees in the University of the Basque Country were also analyzed. Figure 3 shows the competence classification and its relationship to the 'care' and 'provision' concepts.

### 2.1. Gender Distribution in the University System

University degrees are not equally selected by women and men in the Spanish university system. In fact, there is a tendency for women to choose sciences, health sciences, arts and humanities, and

social and juridical sciences. However, as shown in Figure 4, only 25% those enrolled in engineering and architecture degrees in 2015–2016 were women.

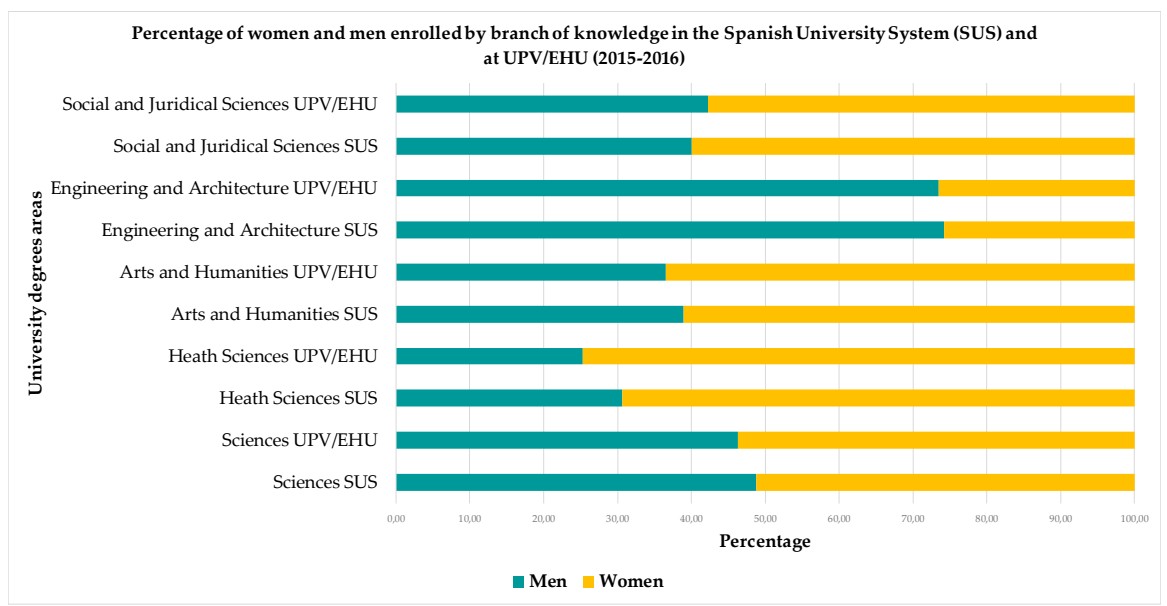

**Figure 4.** Percentage of women and men enrolled by area of knowledge in the Spanish University System [14] and at University of the Basque Country UPV/EHU 2015–2016 [56].

In the University of The Basque Country (UPV-EHU), gender distribution in engineering and architecture also presents different percentages for female student enrolment in the 2015–2016 academic year compared to other disciplines (Figure 4). Moreover, the percentage of female students enrolled is lower in degrees directly linked to the profession of engineer (Figure 5). There are also fewer female students in 'old' degrees than in 'newer' ones less related to the industrial world, which could be considered as horizontal segregation by gender. For example, qualifications such as environmental engineering have 57.84% female students compared to Mechanical Engineering, where only 14.62% female students are recorded, a figure that is very far from the field average.

## 2.2. Professional Competences Analysis

In relation to the professional competences in the mechanical engineering degrees at the University of the Basque Country, Figure 5 shows the competence classification and its relationship to the 'care' and 'provision' concepts defined in Section 2.1. In the analysis of generic and transversal competences for the degree of industrial engineering, it is observed that each competence includes several dimensions. At the same time, each dimension can also be classified attending to instrumental, interpersonal, and systemic characteristics. First, the competences are classified into generic competences (G), which are typical of engineering degrees, and transversal (T) competences, which are common to most university studies. Secondly, competence dimensions are identified. Then, the dimensions are classified as Instrumental Cognitive (IC), Instrumental Technical–Scientific (IT), Instrumental Linguistic (IL), Instrumental Practical (IP), Interpersonal (INT), Systemic Opening Personal (SOP), and Systemic Impact Personal (SIP). Finally, each competence is classified according to its type of action. Actions directly linked to people are related to "care" activities, and actions oriented toward achievements are related to "provision" activities [49].

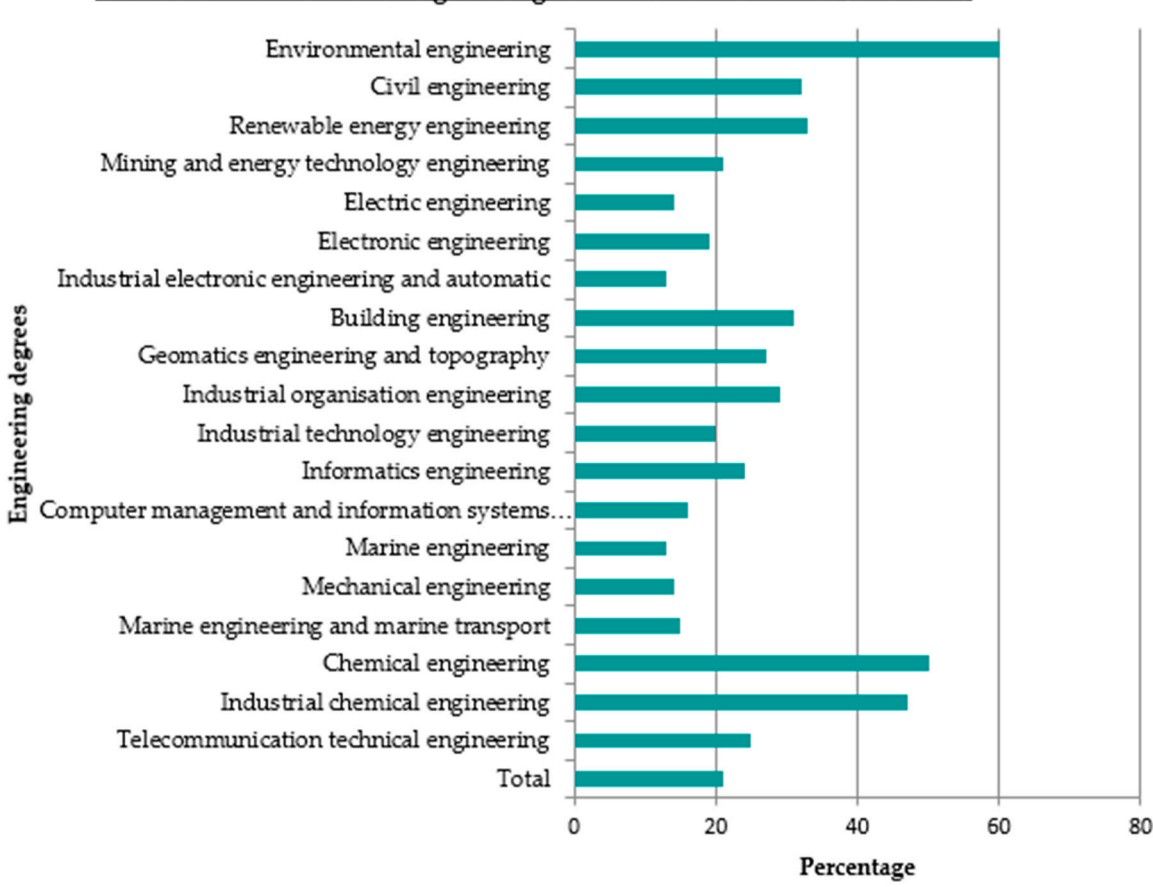

**Figure 5.** Distribution of women in engineering studies at the University of the Basque Country UPV/EHU (2012–2013) [57].

After the analysis, it was observed that 88% of the dimensions for generic competences are oriented toward "provision" activities, while 29% of the dimensions relate to transversal competences. It can be concluded that explicit engineering curricula are focused on "provision" activities, which is translated into a cultural perception of industrial engineering as a male profession. Given that 71% of transversal competences are based on people-centered activities, one way to increase the number of women in engineering may be to promote transversal skills classified as caregiving. Since these are the skills most demanded by companies, they are becoming increasingly important in the new economic and social paradigm. Other actions may focus on factors that intervene in the learning process, beyond the explicit curriculum, such as integrating the knowledge of women and using teaching methodology, focused on people.

If the Mechanical Engineering degree competences, with one of the lowest percentage of female students among engineering studies (Figure 3), are compared to those of chemical-related engineering degrees (Environmental Engineering, Industrial Chemical Engineering and Chemical Engineering), with the highest percentage of female students among engineering studies (Figure 3), these are the obtained results:

Industrial Chemical Engineering degree competences and Mechanical Engineering degree competences are mainly (over 90% of the competences) the same. This is a consequence of the studies plan, taking into account that the subjects studied in the first two years and the fourth year of both degrees are the same. Therefore, in this case, as well as in Mechanical Engineering degrees, 88% of the dimensions for generic competences are oriented toward "provision". In relation to women distribution in engineering studies at the University of the Basque Country UPV/EHU

(2012–2013) (Figure 3), Industrial Chemical Engineering presents the lowest female presence among chemical-related degrees, at 47%.

Chemical Engineering competences present 75% of the dimensions for generic competences are oriented toward "provision". On the other hand, in relation to the distribution of women in engineering studies at the University of the Basque Country UPV/EHU (2012–2013) (Figure 3), Chemical Engineering presents the second lowest female presence among chemical related degrees, at 50%.

In Environmental Engineering, which has the highest percentage of enrolled students (Figure 3) 60%, it can be noticed that only 67% of the generic competencies are of provision. This is due to the fact that greater emphasis is placed on Instrumental Linguistic (IL), Interpersonal (INT) and Systemic Personal Openness (SOP) competencies than in the case of Mechanical Engineering. It is noticeable that Mechanical Engineering and Environmental Engineering differ mainly in the scope of application. Thus, in the case of Mechanical Engineering, the objective is placed on machines and installations (provision), whereas in Environmental Engineering, the aim is correcting the environmental problems generated by industrial activity (care).

After this mechanical and chemical degrees comparison, it should not be forgotten that along with the explicit curriculum, in which competences are included, there are other factors influencing students' choice of university degree. This is, the historical origin of each engineering degree, the image that society has of it, the culture of the educational center in which the degree is taught, the field of application as well as the closest environment to the person. In future works, it will be necessary to analyze the influence of these factors on industrial engineering and on those engineering degrees that present higher percentages of women. (Table 3).

**Table 3.** Professional competence (obtained from engineering degree verified memory documents) analysis of mechanical engineering degrees at the University of the Basque Country UPV/EHU.

| Competence | Type | Classification | Care/Provision |
|---|---|---|---|
| Scientific methodology strategies application: problem situation **analysis** qualitatively and quantitatively. **Propose hypotheses and solutions** *using the models of industrial engineering*, industrial electronic specialty. | G | **IC** *SIP* | **Provision** *Provision* |
| Ability to **analyze and assess the social and environmental impact** of technical solutions. | G | **IC** **INT** | **Provision** **Care** |
| Knowledge, understanding and ability to *apply the necessary legislation* in the exercise of the profession of Industrial Technical Engineer. | G | **ITS** *SIP* | **Provision** *Provision* |
| Ability *to apply* the principles and methods of *quality*. | G | *SIP* | *Provision* |
| **Capacity for organization and planning** in the field of the company, and other institutions and organizations. | G | **IP** | **Provision** |
| Ability to **solve problems** *with initiative*, **decision making**, *creativity*, **critical reasoning and to communicate and transmit knowledge**, skills and abilities in the field of Industrial Engineering. | G | **IC** **INT** *SIP* | **Provision** **Care** *Provision* |
| Ability to write, sign and *develop projects* in the field of industrial engineering for the *construction, reform, repair, conservation, demolition, manufacture, installation, assembly or operation of* structures, mechanical equipment, energy installations, electrical and electronic installations, industrial facilities and processes and manufacturing and automation processes. | G | *SIP* *IP* | *Provision* *Provision* |
| **Ability to handle specifications**, regulations and mandatory standards. | G | **IP** | **Provision** |
| Capacity for *management, of the activities* subject of the engineering projects. | G | *SIP* *IP* | *Provision* *Provision* |

**Table 3.** *Cont.*

| Competence | Type | Classification | Care/Provision |
|---|---|---|---|
| **Knowledge** for the realization of measurements, calculations, valuations, appraisals, surveys, studies, reports, work plans and other analogous works. | G | **IC** | **Provision** |
| **Knowledge in basic** and technological subjects, which enables them to *learn new methods and theories*, and give them *versatility to adapt to new situations*. | G | IC *SOP* | Provision *Care* |
| **Work effectively in a group** integrating skills and knowledge **to make decisions** in the field of industrial engineering, industrial electronic specialty. | T | INT IP | Care Provision |
| **Ability to work in a multilingual and multidisciplinary environment.** | T | INT IL | Care Care |
| Adopt a *responsible attitude, orderly at work and willing to learn* considering the challenge that the necessary continuous training will pose. | T | *SOP* | *Care* |

Data for the participation of women in engineering are very similar in Europe with the exception of Sweden, where the participation of women in STEM fields stands out. This study could be extrapolated within the European Education Area, given that the competences to be acquired by graduates are similar.

## 3. Results and Discussions

After the competence and gender equality diagnosis, with more than 80% of the competence dimensions related to "provision", and because it is not currently possible to update degree competences, actions for the inclusion of a gender perspective in the degree in mechanical engineering are proposed. To this end, the subject content and teaching methodologies have been updated. Thus, we intend to incorporate activities that provide a gender perspective in the context of the subject. Appropriate attitudes towards teaching are also important to develop during teacher education [58]. This work aims to establish a teaching–learning methodology, which promotes active methodologies that allow continuous monitoring and immediate feedback by the students.

- A demonstration of the women who have contributed to the development of the theoretical concepts or practical methods important to the subject.
- The creation of mixed work groups with the added value of diverse points of view and methods of working to highlight the influence that the different aspects of the subject may have on people and, whenever possible, analyze whether this influence affects people differently or specifically.
- The development of a theory class with a question and answer period, including spontaneous questions and answers.
- The inclusion of a practice class or individual or tutorial classes.
- Subject material revisions and the removal of sexist language.
- The inclusion of a gender perspective in the subject materials through exercise statements.

By implementing these measures in the degree curricula, the incorporation of a gender perspective will positively index the acquisition of learning outcomes related to teamwork, the development of proposals and the discussion of ideas, and the ability to evaluate solutions from the perspective of social sustainability, considering people as both the object and subject of study.

These measures will directly influence the academic preparation of students in the training process by providing them with new theoretical and methodological elements for the understanding of social reality. On the other hand, the importance of the class discussion of issues related to a gender perspective will contribute to the development of young people, provide elements for the deconstruction of social gender roles, and facilitate equity and respect for the transmission of different values.

For a more neutral competence description, the analysis carried out in this work needs to be performed. Afterwards, a description of the care and provision competences must be developed equally so that there is a balance between them.

## 4. Conclusions

After our professional competence analysis in engineering studies at the University of the Basque Country (UPV/EHU), the relationship between the masculinization factors present in professional competences and the selection of university studies was determined.

University degrees are not equally selected by women and men. At the University of The Basque Country (UPV-EHU), the female gender distribution in STEM studies is different than that in other disciplines and is even lower than the percentage of female students enrolled in degrees directly linked to the profession of engineer. There are also fewer female students in 'old' degrees than in 'newer' ones not as closely related to the industrial world, which could be considered a type of horizontal segregation by gender.

After the competence analysis, it can be concluded that the explicit engineering curriculum is focused on "provision" factors, which translate into a cultural perception of industrial engineering as a male profession.

However, after mechanical and chemical degrees comparison, it should be stated, that along with the explicit curriculum, in which competences are included, there are other factors influencing students' choice of university degree. This is, the historical origin of each engineering degree, the image that society has of it, the culture of the educational center in which the degree is taught, the field of application as well as the closest environment to the person. In future works, it will be necessary to analyze the influence of these factors on industrial engineering and on those engineering degrees that present higher percentages of women.

In all academic fields, there is great segregation, both horizontally and vertically; the same segregation is manifested intra-engineering. The present actions work towards the incorporation of a gender perspective in the context of the degree in mechanical engineering at The University of the Basque Country (UPV/EHU).

Employing a gender perspective in higher education is essential—on the one hand, because the production of knowledge is not something separate from the gender system. On the other hand, not considering gender in higher education can lead to the professions being biased by the masculinity–femininity dichotomy.

**Author Contributions:** Conceptualization, M.I.T., U.F.-G.; methodology M.I.T., A.C.-O, A.B.; investigation and data curation M.I.T., A.B.; writing—original draft preparation, A.C.-O.; writing—review and editing, A.C.-O, A.B, U.F.-G. and M.I.T.; supervision, M.I.T. All authors have read and agreed to the published version of the manuscript.

**Funding:** This research was funded by PIE (Educational Innovation Projects) program of the University of the Basque Country for their support via 2018–2019 call and grant number [59].

**Acknowledgments:** Thanks are addressed to PIE (Educational Innovation Projects) program of the University of the Basque Country for their support via 2018–2019 call. The funding from the Government of the Basque Country and the University of the Basque Country UPV/EHU through the SAIOTEK (S-PE11UN112) and EHU12/26 research programs, respectively, is gratefully acknowledged.

**Conflicts of Interest:** The authors declare no conflict of interest.

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
