# Peer review of "The Gender Perspective of Professional Competencies in Industrial Engineering Studies"

_sustainability, doi:10.3390/su12072945_

Round 1
Reviewer 1 Report
This work analyses University Degrees' Professional Competences as one of the factors that influences studies selection. Conclusion of the work is very clear: explicit engineering curriculum is translated into a cultural perception of a male profession because most of the generic competences are oriented to masculine activities.
Consequently, authors point out this reason as a relevant factor for career choice.
This point of view is very interesting but this would need further evidence. Authors include a distribution of women in engineering studies (Fig. 4). This can be used to support the conclusions of the work if the competences of an engineering degree with a high percent of women are also analyzed. For example, authors must analyze mechanical engineering in contrast with chemical engineering (or the engineering with the highest percent of women, very difficult to see in the graph). This will help to determine if the competences are so relevant on the choice of studies.
- In addition, the quality of the document could be enhanced by:
- Figure 4 must be revised with some fix as there are less labels than bars.
- Table 3: it seems there is a miss cell content in the specific competence part
- Update data if possible to a more recent academic course.
- Analyze if there is any tendency along years on the percent of women in engineering studies (eg. in mechanical engineering or in engineering and architecture as a macro-area)
- Clarify when engineering refers to mechanical engineering, industrial engineering of any UPV/EHU engineering
- Incluye some proposal for a more neutral competence descriptions
If conclusions of the work are fully supported by a comparison with an engineering degree with a high percentage of women, the outcomes will be very valuable for promoting a deep change on the engineering degrees competences.
Author Response
Dear Reviewer 1,
First of all, we would like to thank your effort and time correcting the paper. In our sincere opinion, your comments and suggestions have helped us to improve the paper.
I attach in next page the point-by-point description of changes made after the comments.
*all the changes introduced in the article are emphasized in red in the text.
Best regards,
Point-by point description of the changes after the reviewing process:
- “This point of view is very interesting but this would need further evidence. Authors include a distribution of women in engineering studies (Fig. 4). This can be used to support the conclusions of the work if the competences of an engineering degree with a high percent of women are also analyzed. For example, authors must analyze mechanical engineering in contrast with chemical engineering (or the engineering with the highest percent of women, very difficult to see in the graph). This will help to determine if the competences are so relevant on the choice of studies.”
According to the reviewer suggestion, obtained results are compared with chemical engineering. In this sense, a new paragraph is added in line 325.
“When comparing Mechanical Engineering with Environmental Engineering, which has the highest percentage of enrolled students, we can see that they differ mainly in the scope of application. Thus, in the case of mechanical engineering the objective is placed on machines and installations (provision), whereas in Environmental Engineering the aim is correcting the environmental problems generated by industrial activity (care).
After analyzing the competencies, we can be notice that in Environmental Engineering only 67% of the generic competencies are of provision. This is due to the fact that greater emphasis is placed on Instrumental Linguistic (IL), Interpersonal (INT) and Systemic Personal Openness (SOP) competencies than in the case of Mechanical Engineering.”
In addition, the quality of the document could be enhanced by:
- Figure 4 must be revised with some fix as there are less labels than bars.
Reviewer suggestion is followed and Figure 4 is corrected. It was a Figure size error, once the figure is enlarged, all the labels can be seen.
- Table 3: it seems there is a miss cell content in the specific competence part
According to reviewer suggestion, Table 3 is corrected and missing cell is completed.
- Update data if possible to a more recent academic course.
Authors agree with reviewer’s opinion. More updated date will be desirable. However, up to now these are the latest data published by the University of the Basque Country in relation to this topic.
- Analyze if there is any tendency along years on the percent of women in engineering studies (eg. in mechanical engineering or in engineering and architecture as a macro-area)
- Reviewer request is followed and a new paragraph with this analysis is added in the introductory section.
- ‘In the Engineering and Architecture studies, in the last 10 years, women presence has suffered a reduction of 2.7 points, from 27.7% in the course 2007/2008 to 25.0% in the academic course 2017/2018 [14]. (Ministry of Education, Culture and Sports) data from the successive reports Data and Figures of the Spanish University System of the Ministry of Education, Culture and Sports.’
http://www.educacionyfp.gob.es/dam/jcr:2af709c9-9532-414e-9bad-c390d32998d4/datos-y-cifras-sue-2018-19.pdf
http://www.educacionyfp.gob.es/servicios-al-ciudadano/estadisticas/universitaria/datos-cifras.html
- Clarify when engineering refers to mechanical engineering, industrial engineering of any UPV/EHU engineering
Reviewer is correct, and, authors have include a clarifying text in order to specify when engineering refers to mechanical engineering, industrial engineering of any UPV/EHU engineering. When engineering term appears alone, without the accompaniment of Mechanical or Industrial ‘surname’ it refers to the set of all engineering degrees.
- Include some proposal for a more neutral competence descriptions
Following reviewer comment, some proposal for neutral competence description are included in the ‘FINDINGS’ section.
Reviewer 2 Report
The paper is super-interesting to me (female professor in an Engineering Faculty). Please consider the following suggestions to further improve the readability:
- Avoid lumped references: try to explain why you quote any single reference. This suggestion mainly apply to the introduction.
- In lines 38-42, as well as in lines 87-92, it is not clear or well explain if you refer to the Spanish situation or if you rather are making general, and not local, statements. In particular the point of the two curricula (which is the plural for curriculum, rather than curriculums) is not clear to me.
- In the introduction, some of the quoted references looks quite old: if still relevant and applicable, maybe that's worth noting.
- Line 68-70: please explain better what you mean by "the authors consider as a male part the engineering school and as a female part the nursing school". What do you mean by "part" there?
- Line 123: "transversal competences, both generic and transversal, ...": the sentence looks odd.
- The taxonomy of the professional competencies is confusing: in section 2.1 they are subdivided into basic / generic / technical; in section 2.2 into generic (instrumental / interpersonal / systemic) and specific, but this subdivision is not matched by Table 3...It is worthy that some additional efforts is dedicated to make the competencies classification clearer.
- Table 1 it is very interesting, but maybe it could be helpful to specify which stereotype/role is masculine and which is feminine.
- Line 162:where does the citation into "quotes" finish?
- Line 195: you mention Specific Competencies: do you mean Professional Specific Competencies? It is not clear, since another alternative categorization is proposed there, different form the previous ones...
- Figure 1: the choice of text in grey makes the written part partly unreadable
- Table 3: The "description" entries could be moved to the text, as a comment/explanation to the Table. Actually, Table 3 could be split in 3 different Tables or reorganized in some more compact solution.
- Section 3 could be skipped, is it mainly a cut-and-paste of sentences already written in the introduction
- Figure 2 and 3 could be compacted in a single figure, with comparative bars for Spanish and UPV/EHU situations.
- Figure 4: many bars are not explained: please provide a label for all the bars, or remove the bars without a label.
- Table 5 is very interesting, but it should be specified where the competences listed in the first column come from
- In ref 16 the year is missing.
As a side remark, I'm curious about the motivation that pushed the authors to investigate the special and delicate issue reported in paper: was that any specific mission or task given by academic authority or anything similar? Maybe a comment on that (basically why you dedicate effort to this analysis) should also be added, as an inspiration to other institutions.
I hope this comments/suggestions help in further improving the excellent paper.
Author Response
REVIEWER #2
Dear Reviewer 2,
First of all, we would like to thank your effort and time correcting the paper. In our sincere opinion, your comments and suggestions have helped us to improve the paper.
I attach in next page the point-by-point description of changes made after the comments.
*all the changes introduced in the article are emphasized in red in the text.
Best regards,
Point-by point description of the changes after the reviewing process:
- Avoid lumped references: try to explain why you quote any single reference. This suggestion mainly apply to the introduction.
Reviewer suggestion is followed and every singles reference is explained instead of using lumped references. A paragraph have been added to better explain all references, and more recent international references has been included.
- In lines 38-42, as well as in lines 87-92, it is not clear or well explain if you refer to the Spanish situation or if you rather are making general, and not local, statements. In particular, the point of the two curricula (which is the plural for curriculum, rather than curriculums) is not clear to me.
Reviewer is right. Lines 38-42 and 68-70 are better explained. In the case of lines 38-42 authors refer to the Spanish situation and legislation. Specified laws are from the Spanish legislation system. In the case of lines 87-92, authors refer to general and worldwide university culture. Moreover, according to reviewer correction, curriculums is corrected by curricula being this last the correct plural form of curriculum.
- In the introduction, some of the quoted references looks quite old: if still relevant and applicable, maybe that's worth noting. A
Reviewer comment is taken into account and references are updated avoiding old and unnecessary ones.
- Line 68-70: please explain better what you mean by "the authors consider as a male part the engineering school and as a female part the nursing school". What do you mean by "part" there?
After reviewer reflection, the authors have rewritten 68-70 sentence. In this case, what reference [20] states is that traditionally, engineering and nursing have been dominated by one gender (in the case of the former by males, and for the latter by females).
- Line 123: "transversal competences, both generic and transversal,” the sentence looks odd.
Author is right. There is an error in the sentences. Is has been rewritten by ‘Initially, competences, both generic and transversal, and competences acquisition methods, are classified into ‘care’ (feminine) or ‘provision’ (masculine).’
- The taxonomy of the professional competencies is confusing: in section 2.1 they are subdivided into basic / generic / technical; in section 2.2 into generic (instrumental / interpersonal / systemic) and specific, but this subdivision is not matched by Table 3...It is worthy that some additional efforts is dedicated to make the competencies classification clearer.
Table 3 appears divided into three levels: transversal competences (corresponding to the basic ones in section 2.1), generic (collected as generic) and specific (corresponding to Technical)
On the other hand, Table 3 distinguishes between instrumental, interpersonal and systemic, as indicated in section 2.2., it deepens by adding a more exhaustive classification depending on the object of reference. To obtain Systemic Opening Personal, Systemic Impact Personal and Instrumental Cognitive, Technical, Linguistic and Practical.
Professional Interest has been added to indicate the vocational character of some traditional professions.
- Table 1 it is very interesting, but maybe it could be helpful to specify which stereotype/role is masculine and which is feminine.
Reviewer is right; it is more helpful if stereotypes/roles are classified as masculine and feminine. So, according to reviewer suggestion, each stereotype has been classified as F (feminine) and M (masculine).
- Line 162: where does the citation into "quotes" finish?
Error in line 162 is corrected and the citation presents now opening and closing quotes.
- Line 195: you mention Specific Competencies: do you mean Professional Specific Competencies? It is not clear, since another alternative categorization is proposed there, different form the previous ones...
The reviewer is right, there is an error in line 195 and where it says ‘Specific Competencies’, authors wanted to say ‘Professional Specific Competencies’. The sentence is corrected in the manuscript.
- Figure 1: the choice of text in grey makes the written part partly unreadable
Thanks to reviewer appreciation, the written part can be easily read now. Author have changes the written part text color from grey to black.
- Table 3: The "description" entries could be moved to the text, as a comment/explanation to the Table. Actually, Table 3 could be split in 3 different Tables or reorganized in some more compact solution.
According to reviewer suggestion, Table 3 has been organized in a more compact solution.
- Section 3 could be skipped, is it mainly a cut-and-paste of sentences already written in the introduction
Section 3 is removed.
- Figure 2 and 3 could be compacted in a single figure, with comparative bars for Spanish and UPV/EHU situations.
Figure 2 and 3 are compacted in a single figure.
- Figure 4: many bars are not explained: please provide a label for all the bars, or remove the bars without a label.
Reviewer suggestion is followed and Figure 4 is corrected. It was a Figure size error, once the figure is enlarged, all the labels can be seen.
- Table 5 is very interesting, but it should be specified where the competences listed in the first column come from
According to reviewer suggestion, where the competences listed in the first column come from. In this case the competences are obtained from the Engineering Degrees Verified Memory documents.
- In ref 16 the year is missing.
Ref 16 is corrected.
- As a side remark, I'm curious about the motivation that pushed the authors to investigate the special and delicate issue reported in paper: was that any specific mission or task given by academic authority or anything similar? Maybe a comment on that (basically why you dedicate effort to this analysis) should also be added, as an inspiration to other institutions.
Following reviewer suggestion, authors motivation in this research paper has been added in the introductory section. Authors belong to the Equality Commission of the Faculty, and, in this case, the motivation comes from the fact that despite the efforts made to increase the number of enrolled women, it is observed that the number of enrolled female students is decreasing. Therefore, it was decided to analyze the culture of mechanical engineering starting with the analysis of the competencies that graduates must acquire.
I hope this comments/suggestions help in further improving the excellent paper.
We would like to thank your effort and time correcting the paper. In our sincere opinion, your comments and suggestions have helped us to improve the paper.
Reviewer 3 Report
The issue this manuscript bringing is interesting, yet quite rare. It discusses the relationship between gender and engineering education. Both of them are among the sustainable development goals, as the epicenter of this journal’s scope. However, there are some of the following suggested to the authors in order to improve the manuscript;
(1)
Given the broad and international readers’ background of this journal, the authors are suggested to provide more portion of references written in English. It is difficult for international readers to follow the ideas presented within the manuscript without digging deeper into referenced papers. In addition, the authors are also suggested to provide the digital object identifier (DOI) of each reference in order to facilitate the readers to read the cited references.
(2)
The authors are suggested to replace the references [2-5] with the more recent papers. Studies in gender perspective are among hot issues in sustainability It says “In the recent years, (…)” however, the most recent paper is from 2006. It is difficult to justify that the issue the authors presenting is among the hot topics if the referenced information is dated nearly 15 years ago.
(3)
The authors are suggested to improve the selection of referenced papers into the newer and more international ones regarding the importance of gender studies in the education sector (last paragraph of section 1, line 113 – 130). The relation is important, however, most of cited papers are too old. Only a paper is from 2018. In addition, selecting papers written in English could improve the readers understanding on the issue this manuscript is presenting.
(4)
The authors need to improve structure of the manuscript into the following order: (1) Introduction, (2) Methods, (3) Results and discussion, (4) Conclusion. Especially on section 4 and 5 of the manuscript, the authors are suggested to rearrange them since the section 4 contains more than section 5. As a reference, the authors are suggested to read the following material http://jultika.oulu.fi/files/isbn9789514293801.pdf regarding the article elements.
(5)
The authors are requested to provide more information regarding the different colors presented in Figure 1 and its placement into different quadrant of each gender. I suppose each color represents either professional competences or formation activities.
(6)
In the section 2.4, the authors are suggested to add more information regarding the Dimension I and II. Line 241 – 244 where the dimensions are mentioned, do not include any information on what the dimensions are, what the purpose of the dimensions, etc. I suppose the dimensions are factors influencing the possibility of a person’s labor insertion. Please provide clearer information about that.
(7)
The authors need to clarify the meaning of Figure 4 since the data are not categorized based on gender and/or the engineering studies. It is difficult to distinguish which data are representing women and/or particular engineering studies. Why the engineering studies consist of more than one data is also not clearly described.
(8)
The authors are suggested to carefully check and recheck the abbreviation used within the manuscript. The consistency in mentioning each abbreviation within the manuscript is important in order to avoid confusion of the readers. In the Table 5, SIP has been mentioned along with SPI.
(9)
The authors are requested to provide more information regarding the selection of different colors presented in Table 5. Why IP Provision is colored both in blue and green, for example. The other explanation regarding the coloring of other classification and care/provision categorization is also needed.
(10)
In the section of results and discussion, the authors are suggested to add more discussion on whether the similar study is also applicable to other countries since the gender perspective is unavoidable from cultural influences. What the condition(s) the other countries need to fulfill in order to apply this kind of study. What the outcome(s) might become if the similar study is carried out on other countries.
Author Response
REVIEWER #3
Dear Reviewer 3,
First of all, we would like to thank your effort and time correcting the paper. In our sincere opinion, your comments and suggestions have helped us to improve the paper.
I attach in next page the point-by-point description of changes made after the comments.
*all the changes introduced in the article are emphasized in red in the text.
Best regards,
Point-by point description of the changes after the reviewing process:
(1)
Given the broad and international readers’ background of this journal, the authors are suggested to provide more portion of references written in English. It is difficult for international readers to follow the ideas presented within the manuscript without digging deeper into referenced papers. In addition, the authors are also suggested to provide the digital object identifier (DOI) of each reference in order to facilitate the readers to read the cited references.
Reviewer comment is taken into account and more international references and DOI identifier is added. ANA
(2)
The authors are suggested to replace the references [2-5] with the more recent papers. Studies in gender perspective are among hot issues in sustainability It says “In the recent years, (…)” however, the most recent paper is from 2006. It is difficult to justify that the issue the authors presenting is among the hot topics if the referenced information is dated nearly 15 years ago.
Reviewer suggestion is followed and more recent international references have been included. Moreover a paragraph has been added to better explain all references, and every single reference is explained instead of using lumped references.
(3)
The authors are suggested to improve the selection of referenced papers into the newer and more international ones regarding the importance of gender studies in the education sector (last paragraph of section 1, line 113 – 130). The relation is important, however, most of cited papers are too old. Only a paper is from 2018. In addition, selecting papers written in English could improve the readers understanding on the issue this manuscript is presenting.
Reviewer comment is taken into account and more international references and updated ones are added.
(4)
The authors need to improve structure of the manuscript into the following order: (1) Introduction, (2) Methods, (3) Results and discussion, (4) Conclusion. Especially on section 4 and 5 of the manuscript, the authors are suggested to rearrange them since the section 4 contains more than section 5. As a reference, the authors are suggested to read the following material http://jultika.oulu.fi/files/isbn9789514293801.pdf regarding the article elements.
According to reviewer suggestion, the structure of the manuscript has changed. Section 3 has been skipped and section 4 has been renamed as Results and discussion. The rest of the sections have been ordered following reviewer suggestion: (1) Introduction, (2) Methods, (3) Results and discussion, (4) Conclusion.
Thanks are addressed to the reviewer for providing an interesting article with tips for writing scientific journal articles.
(5)
The authors are requested to provide more information regarding the different colors presented in Figure 1 and its placement into different quadrant of each gender. I suppose each color represents either professional competences or formation activities.
Authors are sorry to say that colors in figure 1 do not have a special meaning. It is just a way of showing information in a more attractive way.
(6)
In the section 2.4, the authors are suggested to add more information regarding the Dimension I and II. Line 241 – 244 where the dimensions are mentioned, do not include any information on what the dimensions are, what the purpose of the dimensions, etc. I suppose the dimensions are factors influencing the possibility of a person’s labor insertion. Please provide clearer information about that.
Reviewer is right and there is lack of information. A new paragraph is added in order to provide more information in relation to mentioned dimensions.
‘Dimensions pretend to stablish gender dimension in the framework of employability. In this sense, two dimensions are defined, interrelated, to position ourselves in the concept of employability: Dimension I. Gender factor and other structuring factors of personal identity. Dimension II. Deficiencies of the socioeconomic system. Based on these dimensions, we interpret that employability is not a monolithic construct but varies according to the factors.’
(7)
The authors need to clarify the meaning of Figure 4 since the data are not categorized based on gender and/or the engineering studies. It is difficult to distinguish which data are representing women and/or particular engineering studies. Why the engineering studies consist of more than one data is also not clearly described.
There was an error in Figure 4 and it has been corrected. It was a Figure size error, once the figure is enlarged, all the labels can be seen. In this case Figure 4 shows the distribution of women in engineering studies (UPV/EHU 2012-2013) in each engineering degree.
(8)
The authors are suggested to carefully check and recheck the abbreviation used within the manuscript. The consistency in mentioning each abbreviation within the manuscript is important in order to avoid confusion of the readers. In the Table 5, SIP has been mentioned along with SPI.
Abbreviations have been checked and corrected.
(9)
The authors are requested to provide more information regarding the selection of different colors presented in Table 5. Why IP Provision is colored both in blue and green, for example. The other explanation regarding the coloring of other classification and care/provision categorization is also needed.
Tale 5 colors are only a way to identify the competence with the classification and care/provision. However, colors do not have a meaning. In this case in order to avoid confusion cursive, bolt and underlined is used.
(10)
In the section of results and discussion, the authors are suggested to add more discussion on whether the similar study is also applicable to other countries since the gender perspective is unavoidable from cultural influences. What the condition(s) the other countries need to fulfill in order to apply this kind of study. What the outcome(s) might become if the similar study is carried out on other countries.
Following reviewer suggestion, this analysis is added in the 2.2. section.
‘The data for the participation of women in engineering are very similar in Europe with the exception of Sweden, where the participation of women in STEM fields stands out. The study could be extrapolated within the European Education Area, given that the competences to be acquired by graduates are similar.’
http://www.crue.org/Documentos%20compartidos/Publicaciones/Universidad%20Espa%C3%B1ola%20en%20cifras/2018.12.12-Informe%20La%20Universidad%20Espa%C3%B1ola%20en%20Cifras.pdf
Round 2
Reviewer 1 Report
Dear authors,
Thank you very much for taking my comments into account. However, the most important point of my revision needs still further insight. Despite you include in the reply that Quemical Engineering has been considered, in the document, you only refers to Environmental Engineering (lines 320-328). It is very important you include the degrees Chemical engineering and Industrial Chemical Engineering because both represent the desired gender equality. In addtion, the presence of women is significantly higher than the remain degrees.
The following aspects have also to be considered:
- New paragraph starting on line 34 should be revised: Sentence starting on line 35 (According to...) and the next one are not understandable
- Line 59. You must indicate that situation along 10 years is quite stable as there has been very little change
- Line 122. You must justify the choice of mechanical engineering studies
- Figure 1. Is it necessary such figure?. It is very nice but I cannot see the information it provides by colors. Maybe a table or another representation could be useful
- Figure 2 is unreadable
- In paragraph 325, the summary of provision competences in mechanical engineering shoud be also included (88%)
- Table 5. Indicate the degree
Author Response
REVIEWER #1
Dear Reviewer 1,
First of all, we would like to thank your effort and time correcting the paper in round2. In our sincere opinion, your comments and suggestions have helped us to improve the paper.
I attach in next page the point-by-point description of changes made after the comments.
*all the changes introduced in the article are emphasized in red in the text.
Best regards,
Point-by point description of the changes after the reviewing process:
Dear authors,
Thank you very much for taking my comments into account. However, the most important point of my revision needs still further insight. Despite you include in the reply that Quemical Engineering has been considered, in the document, you only refers to Environmental Engineering (lines 320-328). It is very important you include the degrees Chemical engineering and Industrial Chemical Engineering because both represent the desired gender equality. In addtion, the presence of women is significantly higher than the remain degrees.
Reviewer is right and authors have analyzed chemical related degrees. In this case, Environmental Engineering, Industrial Chemical Engineering and Chemical Engineering are also analyzed and compared to Mechanical Engineering.
The following paragraph is included in section 2.2 (line 320).
If Mechanical Engineering Degree competences, with one of the lowest percentage of female students among engineering studies (Figure3), are compared to chemical related degrees (Environmental Engineering, Industrial Chemical Engineering and Chemical Engineering), with the highest percentage of female students among engineering studies (Figure3) this are the obtained results:
Industrial Chemical Engineering Degree competences and Mechanical Engineering Degree competences are mainly (over 90% of the competences) the same. This is a consequence of the studies plan, taking into account that the subjects studied in the first two years and the fourth year of both degrees are the same. Therefore, in this case, as well as in Mechanical Engineering Degrees, 88% of the dimensions for generic competences are oriented toward “provision”. In relation to women distribution in engineering studies at the University of the Basque Country UPV/EHU (2012–2013) (Figure 3), Industrial Chemical Engineering presents the lowest female presence among chemical related degrees, this is, 47%.
Chemical Engineering competences present 75% of the dimensions for generic competences are oriented toward “provision”. On the other hand, in relation to women distribution in engineering studies at the University of the Basque Country UPV/EHU (2012–2013) (Figure 3), Chemical Engineering presents the second lowest female presence among chemical related degrees, this is, 50%.
In Environmental Engineering, which has the highest percentage of enrolled students (Figure3) 60%, it can be noticed that only 67% of the generic competencies are of provision. This is due to the fact that greater emphasis is placed on Instrumental Linguistic (IL), Interpersonal (INT) and Systemic Personal Openness (SOP) competencies than in the case of Mechanical Engineering. It is noticeable that Mechanical Engineering and environmental Engineering differ mainly in the scope of application. Thus, in the case of mechanical engineering the objective is placed on machines and installations (provision), whereas in Environmental Engineering the aim is correcting the environmental problems generated by industrial activity (care).
After this mechanical and chemical degrees comparison, it can be concluded that degress with more percentage of provision competences present lower female presence.
The following aspects have also to be considered:
- New paragraph starting on line 34 should be revised: Sentence starting on line 35 (According to...) and the next one are not understandable
- Line 59. You must indicate that situation along 10 years is quite stable as there has been very little change
- Line 122. You must justify the choice of mechanical engineering studies
- Figure 1. Is it necessary such figure?. It is very nice but I cannot see the information it provides by colors. Maybe a table or another representation could be useful
- Figure 2 is unreadable
- In paragraph 325, the summary of provision competences in mechanical engineering shoud be also included (88%)
- Table 5. Indicate the degree
- New paragraph starting on line 34 should be revised: Sentence starting on line 35 (According to...) and the next one are not understandable.
According to reviewer suggestion, some sentences have been included in this paragraph in order to clarify it.
In recent years, the study of the impact of sexism and gender relations on university education has been a topic of great interest all around the world. Many studies have shown that women remain greatly underrepresented in top positions of many institutions, even at university. As a sample, some recent studies are commented below. In 2016, one study showed that in the Italian academia, especially in top positions, women are underrepresented; it has been found that when a small number of positions is available, females have a significant lower probability of promotion, especially for full professor promotion [2]. According to the U.S Department of education, even if between 2006 and 2017 women presence in doctoral degrees overpassed 50%, fewer women are found in the highest academic rank positions [3]. In 2019, the analysis of the gender pay gap among UK academic economists concluded that job rank is found to be a determinant factor, in other words, women are worse paid because they cannot reach the highest positions [4]. The same conclusion has been reached after analyzing the University of Valencia in Spain in 2015 [5] .
- Line 59. You must indicate that situation along 10 years is quite stable as there has been very little change
Reviewer suggestion is followed and it has been indicated that the situation is maintained stable along this 10 years.
In Engineering and Architecture studies, in the last 10 years, women presence has suffered a reduction of 2.7 points, from 27.7% in the course 2007/2008 to 25.0% in the academic course 2017/2018 [14]. This women presence reduction in this last 10 years has suffered little change and it is maintained stable.
- Line 122. You must justify the choice of mechanical engineering studies
The choice of Mechanical engineering studies is justified.
In this case, the motivation comes from the fact that despite the efforts made to increase the number of enrolled women, it is observed that the number of enrolled female students is decreasing. Therefore, it was decided to analyze the culture of mechanical engineering, since it is one of the degrees with the lowest presence of women and it is the degree in which authors teach, starting with the analysis of the competencies that graduates must acquire.
- Figure 1. Is it necessary such figure?. It is very nice but I cannot see the information it provides by colors. Maybe a table or another representation could be useful
Reviewer suggestion is followed and Figure 1 is changed. Colors are changes and now it is a grey, black and white figure. The text is emphasized in bolt for better reading.
- Figure 2 is unreadable
The reviewer is right; a figure with better quality is embedded. Anyway, we have ready a .tiff version of all the figures to upload to the platform before the publication.
- In paragraph 325, the summary of provision competences in mechanical engineering should be also included (88%)
According to reviewer comment, the summary of provision competences in mechanical engineering is be also included (88%).
Therefore, in this case, as well as in Mechanical Engineering Degrees, 88% of the dimensions for generic competences are oriented toward “provision”.
- Table 5. Indicate the degree
According to the reviewer, the figure caption has been modified in order to clarify this issue.
Table 5. Professional competence (obtained from engineering degree verified memory documents) analysis of mechanical engineering degrees at the University of the Basque Country UPV/EHU

Round 3
Reviewer 1 Report
Dear authors,
I enormously appreciate your effort on improve the quality of the paper and you have had taken all my comments into account. Presently, I am very sorry because I consider the conclusions of the work are not supported by the results.
If we collect the percentage of "male competences" and women in engineering degrees at the UPV/EHU we found:
Mechanical Eng. (88%)-->14.62% women
Industrial Chemical (88%)-->47% women
Chemical Eng. (75%)--> 50% women
Environmental Eng. (60%)-->60% women
At the light of these data, we cannot conclude that the lower the male competences the higher presence of women. 47% of women in Industrial Chemical with a 88% of male competences points out that competences' definition is not critical for the choice of engineering studies.
The present figures are very interesting but conclusions of the study are not supported by them.
These figures highlight the starting point of a new study that goes back to the beginning of Industrial Chemical Eng., Chemical Eng. and Environmental Eng. studies. Because, if you look at these engineering studies you could find the reason for the balance between men and women. According to the results of this paper, the definition of competences is not the reason for the choice of engineering studies. Therefore, you could investigate other reasons, maybe historical reasons, to advance in promote the presence of women in engineering studies. The figures at chemical engineerings are the desired situation.
Author Response
REVIEWER #1
Dear Reviewer 1,
First of all, we would like to thank your effort and time correcting the paper in round3. In our sincere opinion, your comments and suggestions have helped us to improve the paper.
I attach in next page the point-by-point description of changes made after the comments.
*all the changes introduced in the article are emphasized in red in the text.
Best regards,
Point-by point description of the changes after the reviewing proces.
-------------------------------------------------------------------------------------
Reviewer is right and authors have followed reviewer comments. It cannot be concluded that the lower the male competences the higher presence of women. Therefore, a new paragraph, explaining this and changing previous statement, is included in line 352. In addition, this reflection is also included in the conclusion section. Of course, as reviewer mentions, there are other factors influencing students’ choice of university degree. This is, the historical origin of each engineering degree, the image that society has of it, the culture of the educational center in which the degree is taught, the field of application as well as the closest environment to the person.
However, this will require a deeper analysis that will be performed in future works. It will be necessary to analyze the influence of these factors on industrial engineering and on those engineering degrees that present higher percentages of women.
It should not be forgotten, that along with the explicit curriculum, in which competences are included, there are other factors influencing students’ choice of university degree. This is, the historical origin of each engineering degree, the image that society has of it, the culture of the educational center in which the degree is taught, the field of application as well as the closest environment to the person. In future works, it will be necessary to analyze the influence of these factors on industrial engineering and on those engineering degrees that present higher percentages of women.
